# Advances of Non-Ionic Surfactant Vesicles (Niosomes) and Their Application in Drug Delivery

**DOI:** 10.3390/pharmaceutics11020055

**Published:** 2019-01-29

**Authors:** Xuemei Ge, Minyan Wei, Suna He, Wei-En Yuan

**Affiliations:** 1School of Light Industry and Food Engineering, Nanjing Forestry University, Nanjing 210037, China; gexuemei@njfu.edu.cn; 2Engineering Research Center of Cell & Therapeutic Antibody, Ministry of Education, and School of Pharmacy, Shanghai Jiao Tong University, Shanghai 200240, China; weiminyan@sjtu.edu.cn; 3Department of Pharmaceutical Sciences, Medical College, Henan University of Science and Technology, Luoyang 471023, China; hesuna-2008@haust.edu.cn

**Keywords:** niosome, drug delivery, non-ionic surfactant, carrier, stability

## Abstract

Non-Ionic surfactant based vesicles, also known as niosomes, have attracted much attention in pharmaceutical fields due to their excellent behavior in encapsulating both hydrophilic and hydrophobic agents. In recent years, it has been discovered that these vesicles can improve the bioavailability of drugs, and may function as a new strategy for delivering several typical of therapeutic agents, such as chemical drugs, protein drugs and gene materials with low toxicity and desired targeting efficiency. Compared with liposomes, niosomes are much more stable during the formulation process and storage. The required pharmacokinetic properties can be achieved by optimizing components or by surface modification. This novel delivery system is also easy to prepare and scale up with low production costs. In this paper, we summarize the structure, components, formulation methods, quality control of niosome and its applications in chemical drugs, protein drugs and gene delivery.

## 1. Introduction

Nano-carriers such as liposomes, polymersomes, niosomes, micelles and polymer-based vesicles can provide an ideal approach for the delivery of therapeutic agents to target sites in the treatment of diseases [1]. They have attracted attention from researchers because of their advantages, e.g., nanocarriers may prolong the half-life of drugs in serum, avoid uptake by reticulo-endothelial systems (RESs) and reduce non-specific adsorption by optimizing its components or building a multi-functional surface. And they can also protect the drug from degradation in storage and in vivo circulation [2,3]. Nano vesicles are widely used as carriers in delivering (or co-delivering) chemical drugs, protein drugs and gene medicines. Although numerous research works have focused on how to increase the therapeutic efficacy of drugs with low side effects, only a few of them have been approved for clinical use. Our goal in this field is to develop a feasible way to generate therapeutically and clinically useful nano vesicle formulations [4].

Non-ionic surfactant vesicles (Niosomes), which are formulated with non-ionic amphiphiles in certain aqueous solutions by self-assemble technology, were first used in the development of cosmetics. In structure, niosomes are usually multilamerllar or unilamellar vesicles which possess closed bilayers with hydrophilic cavities as both the internal and hydrophobic shells as the outer layers to accommodate the active agents. In recent years, with the development of nanotechnologies in the field of pharmaceutics, more and more studies have focused on niosomes as nanocarriers for drug delivery. Niosomes can be an alternative to liposomes and polymersomes due to their ability to encapsulate different kinds of drugs for the purpose of increasing their stability and efficacy. Unlike other nanoparticles, structurally, liposomes, polymersomes and niosomes have many similarities, and they can all be loaded with both hydrophilic and hydrophobic drugs. Therefore, they could co-deliver both hydrophilic and hydrophobic drugs in one vesicle. Due to excellent biocompatibility and relatively low toxicity, liposomes have attracted much attention, especially after Doxil® was approved by Food and Drug Administration (FDA) and used in clinical trials [5]. Compared with liposomes, niosomes have advantages such as good stability, low cost, easy to be formulated and scaling-up. Niosomes are much more stable because their forming materials, non-ionic surfactants, are more stable than those of lipids both in terms of physical and chemical stability. Also, the PEG on the surface of liposomes which could prolong the half-life after being administrated was limited because the lipid bilayer can maximally tolerate about 5%–6% mol% of PEG, and may cause some stability problems such as the lysis of liposomes at high concentrations. The formulation processing was much easier due to the good stability of the niosomes. And niosomes are much cheaper than liposomes [6,7,8]. Polymersomes could serve as a promising nano carrier, but the membrane-forming material needs lots of synthesis work to obtain the amphipathic block copolymer. The size, zeta potential and in vivo performance of niosomes can be optimized by selecting its components and formulation methods according to the requirements [9]. Some niosomes are commercially available, and clinical trials have indicated the successful application of niosomes as drug carriers [10,11]. Furthermore, Niosomes can be prepared for many kinds of formulations for different clinical uses. For example, one study aiming to investigate novel niosomes based on nano vesicles for the treatment of pulmonary diseases by inhalation completed its Phase I study in 2017. Melatonin niosome oral gel was formulated in order to overcome the problem of absorption and stability. Their pharmacokinetic properties, sleep induction effect and adverse events will be determined in clinical study [12]. Based on these developments and the advantages of niosomes, the structure, components and formulation methods are introduced in this paper and their potential clinical applications are also discussed.

## 2. The Structure and Components of Niosomes

### 2.1. The Structure of the Niosomes

It is important to understand the basic structural units of niosomes, because that may determine which substances can form niosomes and the loading mechanism of drugs for delivery. Similar to the liposomes, niosomes are non-ionic surfactant vesicles with a bilayer structure (Figure 1). Hydrophilic heads are opposite to aqueous solutions and hydrophobic heads are opposite to organic solutions [13]. Bilayer vesicles can be divided into unilamellar and multilamellar vesicles (Figure 1) [12,14]. Multilamellar vesicles are concentric circles constructed by at least 2 bilayer vesicles or a large vesicle embodying one or more small vesicles (Figure 1b,c). Therefore, the particle size of multilamellar vesicles is usually larger than that of unilamellar vesicles. As for unilamellar sorbitan monostearate (C_18_-sorbitan monoester)-cholesterol niosomes, X-ray scattering data showed a bilayer spacing of 15 nm and a thickness of 3.3–3.4 nm. Generally, niosomes are in the sub-micron (colloidal) size range. The particle sizes of small unilamellar vesicles (SUV) were about 10–100 nm, large unilamellar vesicles (LUV) 100–3000 nm, and multi-lamellar vesicles (MLV) greater than 5 μm, while a few "giant" (> 15 um) vesicles have been reported [13,14,15].

### 2.2. The Components of the Niosomes 

A niosome consists of drugs, cholesterol or its derivatives, non-ionic surfactants and, sometimes, ionic amphiphiles. The drugs, both hydrophilic and hydrophobic, can be encapsulated in the niosomes. Hydrophilic drugs are encapsulated in the corresponding core, while hydrophobic drugs are entrapped in the hydrophobic region of the bilayer. The proper amount of cholesterol is added to the niosomes to achieve the most stable formulation due to its interaction with non-ionic surfactants [16]. Only cholesterol cannot form the structure of the bilayer, but it can mix with the bilayer membrane, playing the role of regulating the structure and flexibility of the membrane as a dependable buffer.

In niosomes, non-ionic surfactants are the main ingredient, rather than phospholipids, which is the primary component in liposomes. Non-ionic surfactants used in the niosomes are amphipathic, including terpenoids [17], polysorbates [18], Spans [19], alkyl oxyethylenes (usually from C12 to C18) [20,21] and so on. Squalene, as a member of the terpenoid family, is a natural lipid. It is used to prepare niosomes, with the advantage of enhancing the rigidity and stability of niosome formulations with minimal cytotoxicity in vitro and in vivo [17]. Polysorbate is one of the most important non-ionic surfactants employed in niosome formulations. For example, niosomes containing polysorbate 80 offer excellent properties for gene delivery in formulation and transfection efficiency, because of the polyethylene glycol (PEG) chains present in its structure [17,18]. Similarly, niosomes consisting of polysorbate 20 also display superior performance in vitro. The PEG chains of polysorbate 20 make the surface properties and composition of niosomes similar to that of PEGylated nanoparticles, which do not affect the integrity of the Caco-2-cell monolayer in vitro, allowing the adhesion of nanoparticles to the intestinal epithelium, and activating the transcytosis pathway. Therefore, niosomes consisting of polysorbate 20 can pass in tact through the Caco-2-cell monolayer and then increase the transport of therapeutic agents across intestinal epithelial barrier to obtain a better therapeutic effect [18]. The niosomal carrier (Span 60/Tween 60/cholesterol) can significantly increase the entrapment efficiency of the drugs because of the interaction between the drugs and the acyl chains of Span 60 [19].

Additionally, some charged molecules or ionic amphiphiles, such as dicetyl phosphate (DCP) and phosphatidic acid(negatively charged molecules), stearylamine (SA) and cetylpyridinium chloride (positively charged molecules) are used in the niosomes for three purposes: loading drugs, increasing the efficacy and enhancing stability [12]. For example, the cationic lipid, 2,3-di(tetradecyloxy) propan-1-amine, is combined with non-ionic surfactants to prepare cationic niosomes. The formed cationic niosomes with a positive charge can interact electrostatically with the negatively-charged phosphate groups of the DNA and increase the transfection efficiency [17]. And the cationic niosomes can increase the drug encapsulation efficiency, skin permeation enhancement, and be used to prepare hybrid niosomal complex [22]. Additionally, charged molecules to the bilayer can also increase the stability of niosomes due to a suitable zeta (ζ)-potential. Generally, fully electrostatic stabilization needs a ζ-potential of over +30 mV or below −30 mV, because particles with a high ζ-potential are less likely to aggregate due to electrical repulsion [23,24]. 

## 3. Methods for Formulation and Evaluation of Niosomes

### 3.1. Formation of Niosome by the Proniosomes Method

Proniosomes, also called dry niosomes, are dry-form formulations of the non-ionic surfactant vesicles which can be converted into niosomes after hydration in a short time, and are now widely used in the formulation of niosomes due to their good stability [6,25,26]. Proniosomes consist of a water-soluble carrier coated with non-ionic surfactants, and are easily hydrated into niosomes before usage (Figure 2). This method possesses several advantages such as good physical and chemical stability for long-term storage, convenience for transportation, and ease to scale up [27,28]. And this technology may offer more options for niosomes to be further formulated in different forms, such as tablets and gel [29,30]. Extensive research has also reported that proniosomes could be used successfully in the application of drug delivery through different routes, such as oral, parenteral, dermal, transdermal and ocular [6]. This is the best way to minimize the water content in niosomes in order to improve their stability, and may provide a solution for long-term storage. 

### 3.2. Sonication

Sonication is a conventional method for the preparation of niosomes. This method is easy to operate. The drug solution (in buffer) must simply be added to the proper mixture of non-ionic surfactant at optimized ratio and then sonicated at the determined frequency, temperature and time, to obtain the desired niosomes. This is also a suitable way to control the particle sizes of the niosomes. D. Pando et al. reported that resveratrol niosomes were prepared with an encapsulated rate of 43% by using two-stage technologies: mechanical agitation and sonication. Sonication can decrease the diameters of niosomes with narrow size distribution [31]. But probe sonication involves the use of high levels of energy, and may cause a sudden increase of temperature and the shedding of titanium [7].

### 3.3. Micro Fluidization

Micro fluidization is a new method for the formulation of niosomes, which is based on the jet principle, i.e., by mixing two kinds of fluids such as alcohol and water in microchannels. Niosomes can be formulated with the desired particle sizes and size distribution by optimizing the parameters, such as mixing conditions, surfactants and other materials [32]. The formulation of niosomes by the method of micro-fluidization is widely used. It is reported that Mohammad A. Obeid et al [33]. successfully prepared non-ionic surfactant vesicles for the purpose of delivering therapeutic siRNA into cancer cells using microfludics device NanoAssemble (Benchtop, Precision NanoSystems Inc., Canada). The size of the niosomes was below 60 nm, with relatively narrow distribution and good stability for over 8 weeks at 25 ℃ [33,34]. Due to the advantages of the micro fluidization methods, such as the formation of niosomes with smaller sizes, better reproducibility and ease of formulation, they have been widely used in the formulation of niosomes in recent years. And this method is considered as a promising way for the industrial development of niosomes. 

### 3.4. Thin-Film Hydration Method

Thin film hydration (TFH) is one of the most widely-used methods for the preparation of liposomes. This method could be also used in the formulation of niosomes. It is a simple method which involves dissolving the membrane-forming materials in an organic solvent in a flask. As shown in Figure 3, after removing the organic solvent by vacuum evaporation, a layer of dried thin-film forms inside the flask. The drug is dissolved in aqueous solution such as water or buffer, and then added to hydrate the dry film. It is incubated above the transition temperature of the surfactant in a water bath to form niosomes. Niosomes prepared by TFH method are multilamellar vesicles (MLV). Sometimes, this technique is used together with sonication to acquire niosomes with narrow size distribution. This method is widely used to formulate niosomes loaded with drugs such as insulin, doxorubicin and other extracts [21,35,36].

### 3.5. Reversed Phase Evaporation

As shown in Figure 4, the reverse phase evaporation method involves dissolving the non-ionic surfactant and other additives in an organic solvent. The loaded drug is dissolved in an aqueous solution such as water or PBS and then added to the organic phase to form an emulsion under sonication. The organic solvent is removed by a rotary vacuum evaporator at 40–60 ℃ to form the niosomes [37,38,39]. Compared with the TFH method, vesicles prepared by REV method could yield nanoparticles with uniform size and unilamellar or oligolamellar structures. 

### 3.6. Others

Some other conventional methods are also used for the preparation of niosomes, such as ether injection, micellar solution, trans-membrane pH gradient and the heating method [12,40,41,42,43]. These methods are similar to the formulation methods of liposomes. Compared with the formulation of liposomes, the preparation of niosomes is much easier due to the good stability of the surfactants compared to that of lipids. The membrane contactor method is a suitable way for scaling up. A syringe-pump device is used for the laboratory scale and pilot scales are processed by using a SPG (Shirasu Porous Glass) membrane. The reported size of the niosomes is around 100 nm with narrow size distribution and the encapsulation rate of spironolactone could reach 95.6% (syringe-pump) and 94.7% (membrane conductor), caffeine 9.7% (syringe-pump) and 9.1% (membrane conductor). This new module may provide a promising strategy for scale-up in industry for the production of niosomes [12,44]. The formulation method, components and other properties are summarized in Table 1.

### 3.7. Characterization of Niosomes

Usually, niosomes are evaluated according to their surface morphology, size distribution, zeta potential, drug loading efficiency and stability during the formulation process and storage. These characteristics are very important for niosomes because these factors not only affect the encapsulation rate and stability of the niosomes, but also relate to their performance in vivo. With the development of detection technology, more and more methods are used in the measurement of niosomes. Some commonly-used technologies for the characterization of niosomes are summarized in Table 2.

#### 3.7.1. Sizes and Zeta Potential of Niosomes

Niosomes are spherical in shape and their size may be determined by several techniques, as summarized in Table 2. Their size distribution and polydispersity index are usually determined by laser scattering (DLS) particle size analyzer. To better observe the sharp of the niosomes, SEM, TEM, AFM and STC are used to determine the morphology of the niosomes. As shown in Figure 5, the morphology of the blank niosomes and three kinds of drugs, rifampicin (RIF), isoniazid (INH) and pyrazinamide (PZA)-loaded niosomes were observed by SEM and TEM images. No aggregates were observed and the nature of blank or drug-loaded niosomes was spherical [52]. The self-assembly of niosomes is rarely spontaneous and needs energy as a driving force, such as heating or mechanical stirring [53]. Cryo-SEM could be used for lamellarity determination. And it is reported that large disc-like niosomes, also named discomes, may form with a size range of 11–60 μm when incubated with noisome dispersion with the proper level of solulan C24. These are also applied in drug delivery due to their unique structure. Confocal laser scanning microscopy (CLSM) could be used to identify the difference between niosomes and discomes [7]. Factors which may affect the assembly of niosomes include: (1) Non-ionic surfactant structures (cholesterol is used to avoid aggregation). Hydrophilic lipohpilic balance (HLB) could be used as an indicator of the vesicle forming ability. (2) Membrane additives. (3) Type of encapsulated drug. (4) Surfactant and lipid levels. (5) Hydration temperature, and so on [10]. AFM also could be used to measure the morphology of niosomes, as reported [54]. It was reported that the sizes of the niosomes could range widely, from about 20 nm to 50 μm [55]. Zeta potential is important to the stability of niosomes in solution, and could be measured by using zetasizer, microelectrophoresis and DLS instruments [56]. 

Sizes and zeta potential are very critical to the pharmacokinetics, bio-distribution, toxicity and stability of niosomes. It was found that larger vesicles are likely to accumulate in the lung, liver and spleen with short serum half-lifes after systematic injection. Improper zeta potential may cause the aggregation of the niosomes, and may also invoke some unwanted effects such as toxicity, decreasing targeting efficiency. We hope to formulate vesicles with narrow size distributions and uniform morphologies to control their in vivo distribution.

#### 3.7.2. Encapsulation Efficiency of Niosomes

The encapsulation efficiency of the niosomes represents the capability of vesicles to load therapeutic agents. The definition of niosome encapsulation efficiency is shown in Table 2, and the “total amount” in the formula refers to the amount of drugs used in the formulation. The encapsulation efficiency of niosomes mostly depends on the type of non-ionic surfactant, synthesis method and other agents used in the formulation process, such as cholesterol. It is reported that the encapsulation rate could reach 75%~90% (but that it is commonly in the range of 10%~40%) [57]. For gene materials, we can also label the DNA/RNA with a fluorescent dye such as calcein for florescence measurements to determine the loading efficiency. 

#### 3.7.3. Stability of Niosomes

The stability of the niosomes plays an important role in their formulation development. It is affected by the preparation method, loaded drugs, and types of the membrane forming materials. For their storage, the changes of particle size, zeta potential, morphology and loaded drug leaky rate may be measured to evaluate the stability. To determine the stability of niosomes during circulation, we may incubate these drug-loaded vesicles at 37 °C and in serum (or even in harsh conditions) to mimic situations in vivo [58]. The sizes, zeta potential and leakiness of the loaded drugs in niosomes are measured as a fraction of time to evaluate the stability of these vesicles. The stability of the nano carriers, such as liposomes, polymersomes and some other lipid- or polymer-based particulates remains a big concern for drug delivery. How to improve their stability during formulation/storage and to prevent premature disassembly before reaching the target sites still needs to be addressed. Compared with liposomes, niosomes possess better stability and have the potential for clinical uses.

## 4. The Application of the Niosomes in Chemical Drugs, Protein Drugs and Gene Delivery

Niosomes first emerged in the field of cosmetics, and are now attracting extensive attention as a vesicle delivery system in pharmaceutics. Due to their ability to entrap both hydrophobic and hydrophilic drugs, niosomes are reported as ideal carriers for the delivery of drugs such as doxorubicin, vaccines, insulin, siRNA and so on. Their therapeutic effects are widely applicable (e.g. anti-Alzheimer, anti-cancer, antioxidant, diabetes and antimicrobial) and can be administrated via different methods, such as intravenously, orally and transdermally [59] (Table 3). Here we summarize three types of the drugs which can be encapsulated into niosomes and delivered to target sites. 

### 4.1. Chemical Drugs

For nano-vesicle-based delivery systems, niosomes can be used as an alternative to liposomes and polymersomes for chemical drug delivery. They possess both a hydrophilic cavity and hydrophobic shell, and are suitable for chemical drug loading. They can also provide a way for the co-delivery of two different kinds of drugs to achieve the desired therapeutic effects. As with liposomes and polymersomes, niosomes have some advantages such as biocompatibility, low toxicity, biodegradability, etc. Furthermore, their good stability, low cost and ease of storage make them an alternative to liposomes. Niosomes were developed as carriers of chemical drugs for the treatment of various diseases such as cancer, diabetes, inflammation and so on.

One application of niosomes in delivering chemical drugs is the use of this formulation to improve oral bioavailability. Carvedilol is a kind of clinical drug that is widely used in the treatment of congestive heart failure and coronary artery diseases. But its systemic availability is limited due to the first-pass metabolism and short half-life after administrated. Numerous studies have worked on ways of developing new formulations to improve the bioavailability of carvedilol. Niosome is considered as one solution, because it can protect the loaded drug from degradation, control the releasing profiles by optimizing its components and avoid first-pass metabolism [66]. It was reported that carvedilol niosomes can be prepared by a film hydration method with a minimal size of 167 nm (PDI 0.6) and highest encapsulation rate 77.7% in different formulations. And it has been proved that the release of all formulations could reach almost 100% with no significant difference after 20 h. The best stability of the vesicle was observed in two different kinds of formulations (C_50_S60_25_T60_25_ and C_40_S60_30_T60_30_) by determining the sizes changes [67]. All these results show that niosomes might be developed and used as a nano-carriers for the oral delivery of therapeutic agents to improve their bioavailability. Niosomes can be also used as carriers for the delivery of chemical drugs for the treatment of cancer due to their smaller size, offering a possibility of enhanced permeability and retention in tumor tissue [68]. Niosomes are also incorporated into hydrogels and chitosan/glyceryl monooleate (CH/GMO) as a pH sensitive formulation for the efficient treatment of cancer [69].

### 4.2. Protein and Peptide Drugs

Protein and peptides such as insulin and bacitracin may function as important therapeutic agents for the treatment of diseases. But their clinical application is hindered due to poor bioavailability, instability during storage and after administration, and also some side effects during the application. To overcome these problems, niosomes may serve as good carriers for the delivery of various protein and peptide drugs, and also show good performance in vaccine formulation and application [70,71]. 

The oral delivery of protein and peptide drugs is still a challenge for macro-biological molecules. For decades, the non-invasive administration of insulin formulations has attracted extensive attention and much research. But until now, no truly non-invasive drug formulation is available. It is reported that niosome was investigated for the delivery of insulin via the parenteral and vaginal routes, and that it showed a good ability to protect insulin from degradation [72,73]. Pardakhty investigated a method for the formulation of insulin niosomes (composed of polyoxyethylene alkyl ether surfactant Brij 52 and Brij 92 or sorbitan monistearate Span 60 and cholesterol) and studied the pharmacokinetic properties of the insulin encapsulated in niosomes in diabetic rats [74]. The insulin niosome was administrated orally and its release profile was measured in simulated intestinal fluid (SIF) and simulated gastric fluid (SCF). The results showed that niosomes could protect insulin from degradation. The insulin niosomes could reduce the blood sugar as expected, and the relative bioavailabilities (F) were 1.88 ± 0.43, 1.46 ± 0.43 and 1.12 ± 0.57 (%) respectively for three different formulations Brij 92, Span 60 and Brij 52 orally [74]. Another example for the successful delivery of protein/peptide drug is H. Yoshida’s investigation into the possibility of peroral administration of 9-desglycinamide 8-arginine vasopressin (DGAVP) by choosing stable noisome-forming materials such as polyoxyethylenealkylethers. In vitro intestinal absorption of encapsulated DGAVP in niosomes was performed using an intestinal loop model to mimic an in vivo situation. The results showed that the DGAVP entrapped in niosomes could achieve relatively high concentrations in the acceptor phase of the rat intestinal lumen compared with a DGAVP solution and DGAVP in the presence of empty niosomes after 120 min [75].

Another application of niosomes is their usage in vaccine formulations. It is known that vaccines are a powerful tool to prevent and eradicate diseases, but their safety and efficacy are still big problems for their application. Protein subunit vaccines, which have been proven to be much safer than live organism-based vaccines, may provide an alternative for vaccine development [76]. Anil Vangala and colleagues developed a non-ionic surfactant involving a nano-vector which aimed to improve the physical stability of a dimethyldioctadecylammonium vesicular adjuvant system. The non-ionic surfactants, such as 1-monopalmitoyl glycerol (MP), cholesterol (Chol) and trehalose 6,6’–dibehenate (TDB) were added to investigate the changes in stability by measuring the changes of vesicle size and zeta-potential in two different temperatures. The results showed that the sizes of MP-Chol-DDA-TDB and MP-Chol-DDA were slightly changed at 25 °C. The efficacy of the formed vaccine formulation was also investigated in this study, and the adjuvant activity was determined in mice against three subunit antigens. Both MP- and DDA-based vesicle formulations could induce antibody responses [24]. These results could provide a way for the development of noisome-based vaccine formulations for disease prevention and therapy. 

### 4.3. Gene Delivery

Gene therapy, as a new modality for the treatment of diseases, has emerged as a powerful tool in recent years. But delivery remains a problem for clinical applications. Non-viral gene carriers which are mainly based on polymers and lipids are employed as two approaches for the delivery of gene materials. Lipoplex is a widely-used gene delivery carrier which may cause toxicity and non-specific attachment during the circulation in vivo [77,78]. 

Instead, niosomes have been widely used as oligonucleotide carriers for the treatment of many kinds of diseases in reported studies. They can be used for the delivery of gene materials due to some advantages such as good chemical and physical stability, relatively smaller sizes, etc. G. Puras reported a method to deliver pCMSEGFP plasmid to the retina using niosomes. They formulated the niosomes based on cationic lipid 2,3-di(teradecyloxy)propan-1-amine, aqualene and polysorbate 80 by a method of solvent emulsification-evaporation. The results proved that niosomes could protect DNA from degradation and help the gene materials to enter cells [17]. For DNA vaccines, niosomes can also be used as vectors and provide a simple, stable and cost effective solution compared with liposomes. S.P. Vyas and colleagues found that by using niosome as gene carriers, DNA encoding hepatitis B surface antigens (HBsAgs) could be encapsulated and invoked an immune-response to produce serum antibodies and endogenous cytokines comparable to that of intramuscular recombinant HBsAgs and topical liposomes [79]. Niosomes can also serve as a delivery system for targeting stem cells [80,81]. A study of niosomes proved that they could function as a platform for the delivery of RNAs to human mesenchymal stem cells for the purpose of promoting cell differential. The design of the niosomes for intracellular delivery of siRNA/miRNA and labelling is shown in Figure 6. These cationic niosomes consist of Span 80, DOTA, and PEGylated lipid (TPGS). RNAs are complexed with niosomes in the proper ratio and the surface charge is around 29.5 mV by DLS measurement, which can result in specific gene silencing in hMSCs [80]. 

### 4.4. The In Vivo Stability, Biodistribution and Formation of Protein Corona of Niosomes 

The in vivo stability of the nano-carrier is an important factor for their delivery efficiency. As described, niosomes were much more stable due to the good chemical and physical stability of their forming materials. So, this may enhance their stability before targeting during in vivo circulation. Their stability is also affected by their surface characteristics such as zeta potential. It is known that positively-charged nanoparticles may cause non-specific adsorption and accumulate in some organs such as the liver. Some experiments were carried out to mimic in vivo situations and determine the performance of the niosomes in biological environments by surface charge measurements, zeta potential, gel electrophoresis and ELISA [82]. Niosomes could prolong the half life during circulation, reduce capture by the liver and improve the uptake of the loaded drugs [83,84]. And it is reported that niosomes could also increase the uptake of methotrexate(MTX) into the brain due to the possibility that niosome components could permeate the blood brain barrier [85]. Animal experiments were also performed to investigate the pharmacokinetics of niosomes. In one study regarding niosome distribution and anti-tumor activity, it was found that the area under the plasma level-time curve increased 6 fold when doxorubicin niosomes were administered, compared to doxorubicin solution, and the area under the tumor level-time curve also increased significantly [84].

Nanoparticles can be administrated via different ways such as inhalation, subcutaneous injection and intravenous injection. They are immediately exposed to high levels of protein in the bloodstream and rapidly adsorb proteins on their surface to form complex protein coronas, as illustrated in Figure 7. Nanoparticle-protein coronas could bind with high affinity (stable complex with long lifetime, known as hard corona) or low affinity (dynamic with shorter life time, soft corona). The formed protein coronas may cause protein misfolding and aggregation, and invoke an immune response. In the mean time, protein coronas could mask or block the functional groups on the surface of the nanoparticle. Some of the corona may cause a loss of function due to the changing of orientation or displacement of target molecules on the surface of the nanoparticles [86,87,88]. This may affect the behavior of nanoparticles in biological systems. So, it is crucial to understand the rational of how niosomes interact with biological components for their further development. The forming materials, size and surface properties may be key factors to determining the formation of coronas [89]. The investigation of protein coronas on niosomes could help us to better evaluate toxicity and help in the application of niosomes in clinical trials.

## 5. Conclusions

Niosomes may function as a fantastic nano-vesicle delivery platform and provide a promising method for the delivery of chemical drugs, protein drugs and gene materials for the purpose of disease prevention and treatment. Compared with liposomes, they have some advantages, such as good chemical and physical stability, low cost and easy formulation. They may prove to be an alternative to liposomes and attract extensive attention in the field of pharmaceutics. More work may be undertaken in the fields below to yield more information for niosome development: (1) Development of multifunctional noisome-based target delivery systems by surface modification. Target molecules could be selected and immobilized on the surface of the niosomes. (2) Studies to investigate the toxicity, especially long term trials to evaluate the safety for their clinical use. (3) From bench to clinical application development and scale up studies to investigate the applications of niosomes in industrialization.

## Figures and Tables

**Figure 1 pharmaceutics-11-00055-f001:**
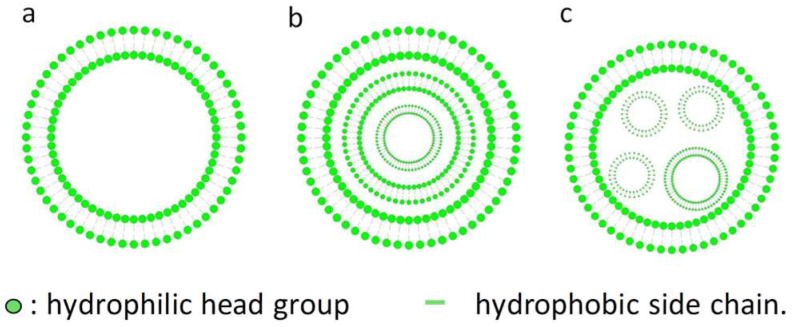
Schematic structures of non-ionic surfactant vesicle. (**a**) unilamellar vesicle, (**b**,**c**). multi-lamellar vesicle.

**Figure 2 pharmaceutics-11-00055-f002:**
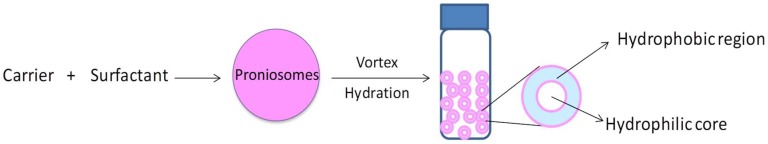
Formation of niosomes by proniosomes methods.

**Figure 3 pharmaceutics-11-00055-f003:**
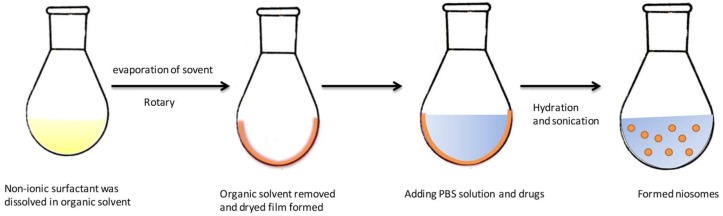
Preparation of niosomes by the thin-film hydration method. Reproduced with permission from [13], published by Elsevier, 2014.

**Figure 4 pharmaceutics-11-00055-f004:**
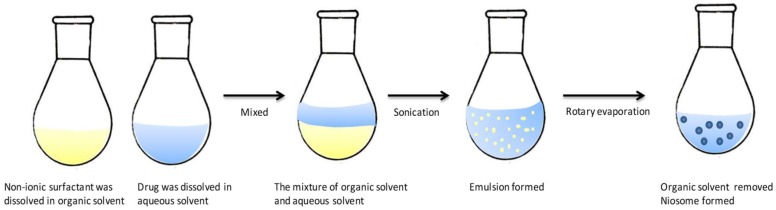
Preparation of niosomes by the reversed phase evaporation method. Reproduced with permission from [13], published by Elsevier, 2014.

**Figure 5 pharmaceutics-11-00055-f005:**
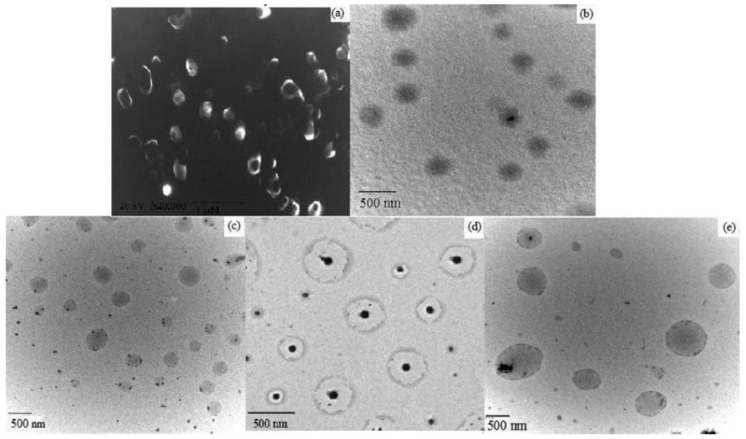
(**a**) SEM image of blank niosomes. TEM images of (**b**) black, (**c**) RIF, (**d**) INH and (**e**) PZA encapsulated niosomes. Reproduced with permission from [52], published by Elsevier, 2011.

**Figure 6 pharmaceutics-11-00055-f006:**
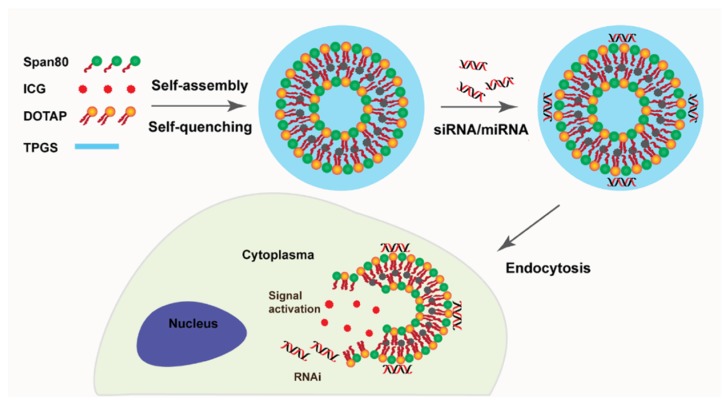
The design of theranostic niosomes for intracellular delivery of siRNA/miRNA and labelling of cells upon dequenching, reproduced with permission from [80], published by American Chemical Society, 2018.

**Figure 7 pharmaceutics-11-00055-f007:**
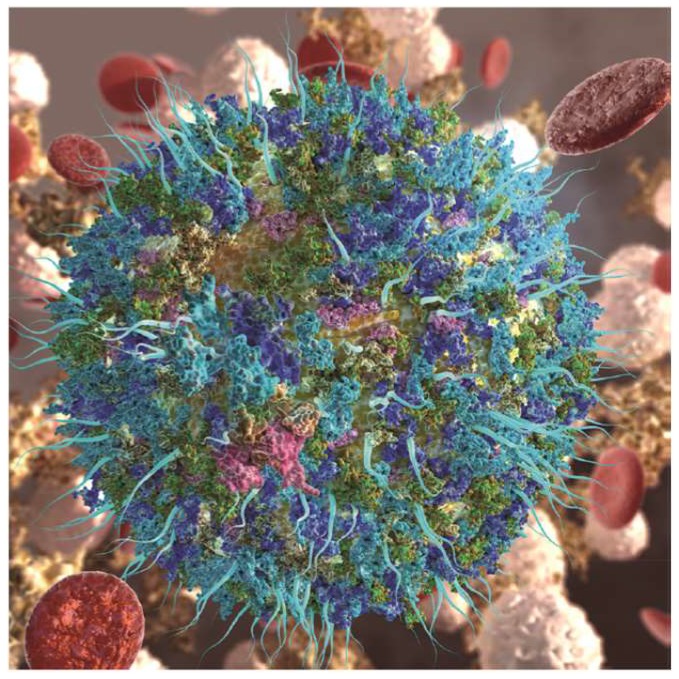
A nanoparticle gains a new biological identify upon its dynamic interactions with biological fluids, giving rise to a protein corona (shown as adsorbed green, blue, and cyan globules), which consequently influences drug delivery and the targeting of functionalized nanoparticles (illustrated as aqua blue fibrils), reproduced with permission from [86], published by American Chemical Society, 2017.

**Table 1 pharmaceutics-11-00055-t001:** Formulation method of niosomes.

Formulation Method	Components	Structures Size (nm)	Zeta Potential (mV)	Encapsulate Rate (%)	Application
Proniosomes	Span 60	Unilamellar 4400 ± 210	/	99.2 ± 5.1	Analgesic, anti-inflammation [45]
	Sugar esters	1620 ± 170	/	98.74 ± 0.51	Disorders cerebrovascular/cerebral degenerative diseases [46]
	Span 40 and chol or DCP or lecithin	multi-lamellar more than 20 μm	/	16.7 ± 1.01 (highest)	antihistaminic [47]
Sonication	Span 60 cholesterol	Multi-lamellar 35.77	(probably higher zeta potential)	29.2 %	anti-inflammation [48]
Micro fluidization	Monopalmitin glycerol cholesterol dicetyl phosphate	From 60.96 ± 0.36 to 168.40 ± 2.26 in different buffer	From −76.83 ± 0.81 to −30.63 ± 2.06 in different buffer	/	[34]
Thin-film hydration method (TFH)	Polyoxyethylene alkyl ethers or sorbitan monoesters	From 214 to 1368	From −26.73 to −41.31	79.8 ± 3.5% (Span 40) 76.56 ± 2.1% (Span 20)	Treatment of Androgenetic alopecia [49]
	Span 60 and cholesterol	5000 ± 1500	/	2.05 ± 0.043/210 Entrapment level (mg)/total lipid (mg)	Treatment of psoriasis [50]
Reversed phase evaporation (REV)	Span 40 or Span 60	3460, 3610	/	26.27% ± 1.96 (highest)	Treatment of glaucoma [51]

**Table 2 pharmaceutics-11-00055-t002:** Methods for characterization of niosomes.

Niosome Parameter	Measurement
Size	DLS, SEM, AFM, STM, CLS
ζ-potential	DLS, Electrophoretic mobility
Encapsulation efficiency (%)	$\mathrm{Encapsulation}\;\mathrm{efficiency}\;=\frac{\;\mathrm{Encapsulated}\;\mathrm{amount}}{\mathrm{total}\;\mathrm{amount}}\times 100\text{\%}$ The amount of the loaded drug is determined by HPLC, UV/VIS, Fluorescence
Stability	DLS (determine size and zeta potential in 37 °C, or in serum to mimic the in vivo situation), Leaky of the loaded drugs

Abbreviations: DLS (Diameter laser scatter), SEM (scanning electron microscope), AFM (Atomic Force Microscope), STM (Scanning Tunneling Microscope), HPLC (High Performance Liquid Chromatography).

**Table 3 pharmaceutics-11-00055-t003:** The application of niosomes in delivering of drugs.

	Surfactant	Formulation Method	Loaded Drug	Encapsulation Rate (%)	Administrated	Application	**Ref**
1	Pluronic L64	REV	Doxonrubicin	38.73 ± 1.58	/(cell level)	Anti-caner	[21]
2	Span 60 Tween 60	REV	Ellagic acid	38.73 ± 1.58	Transdermal	Antioxidant	[60]
3	Tween 20	TFH and Sonication	Curcumin	74.5 ± 3.2	/	Anti-cancer Antioxidant Anti-inflammatory	[61]
4	Tween61	TFH and Sonication	Tyrosinase Plasmid (pMEL34)	150 µg/16 mg of niosomal compositions	Transdermal (*in vitro*)	Treatment of vitiligo	[62]
5	Polysorbate Cationic lipid	REV	pUNO1-hBMP-7 plasmid	/	/	Bone regeneration	[63]
6	Cationic lipid Tween 80 squalene	REV	pCMSEGFP	/	Ocular	Gene delivery	[17]
7	Polyoxyethylene alkyl ethers	THF	Insulin	/	Oral	Diabetes	[35]
8	N-Palmitoyl-glucosamine Span 60	Sonication	Vasoactive Intestinal peptide	24.07 ± 0.83	Intravenous administration	Anti-inflammatory Immunomodulatory neurological Disorders and so on	[64]
9	Monopalmitoyl glycerol	Melt method	H3N2 antigen (Radio-labellin)	/	Oral Intramuscular	Flu	[65]

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
