# Peer review of "Advances of Non-Ionic Surfactant Vesicles (Niosomes) and Their Application in Drug Delivery"

_pharmaceutics, 2019, doi:10.3390/pharmaceutics11020055_

Round 1
Reviewer 1 Report
This review article titled "Advances of Non-Ionic Surfactant Vesicles 3 (Niosomes) and Their Application in Drug Delivery" provides very superficial informations. This review article must be explanded each and every sections to achieve a meaningful article that will be readable well by scientific society. 1. Show the Figure evidences by TEM/SEM/AFM images that are published in literature and what are reasons to get such structures?. 2. Section 2.2. requires generation of table which provides number of components and nisosome structures, particle size, and other physico-chemical properties and their use in drug delivery with examples. 3. Section 3 must be evalborated and each method must discuss in detail and a table with at least 5-10 examples. and finally which method is best for what? 4. Characterization of nisomes is nothing much presented, Indeed any nanoparticles characterize with same technique? Here authors should provide details with some examples? why these techniques needed compared to other nanoparticle characterization? 5. Applications also need to be elaborated in each section with schemes and published literature figures and example tables.
Author Response
This review article titled "Advances of Non-Ionic Surfactant Vesicles (Niosomes) and Their Application in Drug Delivery" provides very superficial informations. This review article must be explanded each and every sections to achieve a meaningful article that will be readable well by scientific society.
1. Show the Figure evidences by TEM/SEM/AFM images that are published in literature and what are reasons to get such structures?
Thank you so much for your suggestion. We added published TEM/SEM/AFM images in the manuscript(Figure 5). The reasons to get such structures were added in Page 7 line 218.
2. Section 2.2. requires generation of table which provides number of components and niosome structures, particle size, and other physico-chemical properties and their use in drug delivery with examples.
Thank you so much for your suggestion. We added table 1 in section 2.2 in page 6 line 204 to provide more information regarding components and niosome structures, particle size, physic-chemical properties and their uses.
3. Section 3 must be elaborated and each method must discuss in detail and a table with at least 5-10 examples. and finally which method is best for what?
Thank you so much for your suggestion. We also added it in table 1. in section 2.2 to give more information about this part in page 6 line 204 and also added more details about advantages or disadvantages of their applications in the manuscript in Page 4 (3. Method for formulation and evaluation of niosomes).
4. Characterization of niosomes is nothing much presented, Indeed any nanoparticles characterize with same technique? Here authors should provide details with some examples? Why these techniques needed compared to other nanoparticle characterization?
Thank you so much for your suggestion. The method for characterization of niosomes is mostly the same as other nanoparticle, such as liposome, polymersomes in technique. More details regarding the size, Zeta potential, stability and encapsulate rate were added in manuscript in this part. Some examples were given in page 7(3.7 Characterization of niosomes)..
5. Applications also need to be elaborated in each section with schemes and published literature figures and example tables.
Thank you so much for your suggestion. We added example tables (Table 3) in this part in page 9. And highlight this part with yellow background.

Reviewer 2 Report
The current manuscript mainly reviews the recent advances of non-ionic surfactant vesicles (niosomes) and their applications in drug delivery. The topic of the review seems interesting. However, this manuscript can be improved further in order to make this review stronger and tighter. My specific comments are given below.
1. The authors should provide more literature illustrations for each sections.
2. It would be nice to summarize the literature reported physico-chemical properties of niosomes (size, zeta-potential, method of fabrication and drug encapsulation efficiency).
3. What are the advantages of niosomes over conventional liposomes and other delivery systems? These should be clearly pointed out.
4. In addition to the applications of gene and drug delivery, it would also be nice to present a section focusing on the biodistribution (pharmacokinetics) of niosomes in the in vivo.
5. The stability of niosomes in the in vivo needs to be elaborated and the formation of protein corona needs to be discussed.
6. There are some typos in the manuscript. Need a careful editing. For example, please correct the spelling of “fluorescence” in pg 5, line 188.
Author Response
Comments and Suggestions for Authors The current manuscript mainly reviews the recent advances of non-ionic surfactant vesicles (niosomes) and their applications in drug delivery. The topic of the review seems interesting. However, this manuscript can be improved further in order to make this review stronger and tighter. My specific comments are given below.
1. The authors should provide more literature illustrations for each sections.
Thank you so much for your suggestion. We have added more literature illustrations and added three figures in our manuscript.
2. It would be nice to summarize the literature reported physico-chemical properties of niosomes (size, zeta-potential, method of fabrication and drug encapsulation efficiency).
Thank you so much for your suggestion. We have added more details in size, zeta-potential, stability and encapsulation efficiency in 3.7 characterization of niosomes in Page 6, line 197. And more methods of fabrication were added in 3.4 Thin-film hydration method (Page 5 Line 166) and 3.5 Reversed phase evaporation (Page 5 Line 178). And two figures were added in this part.
3. What are the advantages of niosomes over conventional liposomes and other delivery systems? These should be clearly pointed out.
Thank you very much for your suggestion. We added more details regarding the advantages of the niosome in manuscript in Page 2, line 49 and highlighted with yellow background.
4. In addition to the applications of gene and drug delivery, it would also be nice to present a section focusing on the biodistribution (pharmacokinetics) of niosomes in the in vivo.
Thank you so much for your suggestion. We have added one section 4.4 The in vivo stability, biodistribution and formation of protein corona of niosomes focusing on the biodistribution(pharmacokinetics) of niosomes. We highlighted this section with yellow in manuscript in Page 9, line 350.
5. The stability of niosomes in the in vivo needs to be elaborated and the formation of protein corona needs to be discussed.
Thank you so much for your suggestion. We have added one section 4.4 The in vivo stability, biodistribution and formation of protein corona of niosomes focusing on the stability and formation of protein corona of niosomes. We highlighted this section with yellow in manuscript in Page 9, line 350. And one figure added in this section.
6. There are some typos in the manuscript. Need a careful editing. For example, please correct the spelling of “fluorescence” in pg 5, line 188.
Thank you so much for your suggestion. We have revised this spelling mistake in line 188 and the typos in manuscript. The revised parts were highlighted with yellow and listed as below:
a. Page 6 Line 188( Line 229): fluorescence.
b. Page 1. Line 15: “has attracted much attention…” changed to “have attracted much attention”
c. Page 1. Line 16: “due to its…”changed to “due to the…”
d. Page 1. Line 30: “the target sites…” changed to “target sites…”
e. Page 3. Line 106: “Polysorbates” changed to “Polysorbate”
f. Page 3. Line 121: “aimes” changed to “aims”
g. Page 3. Line 124: “whith” changed to “with”
h. Page 4. Line 144: “Figure 2. Formation of niosomes by the proniosomes method” changed to “Figure 2. Formation of niosomes by proniosomes method”
i. Page 5. Line 193: “are” changed to “is”
j. Page 6. Line 205: “Methods for characterization of the niosomes” changed to “Methods for characterization of niosomes”
k. Page 6, Line 217: “in lung、 liver and…” changed to “in lung, liver and…”
l. Page 7. Line 237: “the stability of the niosomes” changed to “the stability of niosomes”
m. Page 7. Line 243: “before the target sites is a big concern” changed to “before the target sites still need to be addressed”
n. Page 7. Line 244: “niosomes possesse…” changed to “niosomes possess…”
o. Page 8. Line 284: “its clinical application…” changed to “their clinical application…”
p. Page 8. Line 286: “To overcome these problems. Niosomes may serve as” changed to “To overcome these problems ,niosome may serve as”
q. Page 8. Line 325: “in recent years butthe delivery remains aproblem for its clinical application” changed to “in recent years. But the delivery remains a problem for its clinical application”
r. Page 8 Line 326: “Non-viral gene carriers which are mainly based on polymer and lipids…” changed to “Non-viral gene carriers which mainly based on polymer and lipids…”
s. Page 9 Line 335: “help the gene materials to entry the cells…” changed to “help the gene materials to entry cells…”
t. Page 9 Line 346: “resulte” changed to “result”
u. Page 10 Line 391: “ease” changed to “easy” ; “It emerges” changed to “It may emerge”
v. Page 10 Line 395: One sentence added “Target molecule could be selected and immobilized on the surface of the niosomes”

Round 2
Reviewer 1 Report
All suggestions were implemented.
However, ref 22 is not complere....
Author Response
Dear Felicia Chen, Assistant Editor, and Reviewers,
Thank you very much for your letter and the comments from the referees about our paper submitted to pharmaceutics (Manuscript ID: pharmaceutics-421355R1)! We have checked the manuscript and revised it according to the comments. We submit here the revised manuscript as well as a list of changes.
If you have any question about this paper, please don’t hesitate to let me know.
Sincerely yours,
Yuan

Reviewer 2 Report
The authors have addressed all my revisions. I recommend this paper for publication.
Author Response

(The authors gave the same response as above.)
